# FINDING THE NUMBER OF CLUSTERS IN A GRAPH: A NEARLY-LINEAR TIME ALGORITHM

## ABSTRACT

Given an undirected graph $G$ with the normalised adjacency matrix $N_G$, the well-known *eigen-gap heuristic* for clustering asserts that $G$ has $k$ clusters if there is a large gap between the $k$th and $(k+1)$th largest eigenvalues of $N_G$. Although this heuristic is well-supported in spectral graph theory and widely applied in practice, determining $k$ often relies on computing the eigenvalues of $N_G$ with high time complexity. This paper addresses this key problem in graph clustering, and shows that the number of clusters $k$ implied by the eigen-gap heuristic can be computed in nearly-linear time.

## 1 INTRODUCTION

Graph clustering is a fundamental problem in unsupervised learning with wide-ranging applications across computer science and other scientific disciplines. Among the various techniques for solving graph clustering problems, spectral clustering is probably the easiest to implement, and has been widely applied in practice. Given any graph $G = (V_G, E_G)$ and parameter $k \in \mathbb{N}$ as input, spectral clustering first computes the top $k$ eigenvectors of the normalised adjacency matrix of $G$ and uses these to embed the vertices of $G$ to points in $\mathbb{R}^k$. Afterwards, it applies $k$-means on the embedded points to partition $V_G$ into $k$ clusters (Ng et al., 2001).

Since spectral clustering requires the value of $k$ as input, which is often unknown for real-world applications, to determine $k$ we usually rely on the eigen-gap heuristic: the correct number of clusters corresponds to the smallest $k$ for which there's a clear gap between the $k$th and $(k+1)$th largest eigenvalue of the normalised adjacency matrix of $G$ (von Luxburg, 2007). While this heuristic can be well explained in theory (Davis & Kahan, 1970; Lee et al., 2014), applying it in practice can be computationally expensive and requires computing all eigenvalues, particularly for large values of $k$. Given that spectral clustering itself runs in nearly-linear time (Peng et al., 2017), determining $k$ is the main computational bottleneck in the overall spectral clustering framework. This paper studies this fundamental problem in spectral clustering, and presents the following result:

**Theorem 1** (Informal Statement of Theorem 6). *Let $G = (V, E)$ be an undirected graph with $n$ vertices and $m$ edges as input, and assume the $G$ consists of $k$ well-defined clusters. Then, there is a randomised algorithm that runs in $\widetilde{O}(m)$ time[1] and with probability at least $1 - o(1)$ returns the value of $k$.*

To the best of our knowledge, Theorem 1 represents the first nearly-linear time algorithm that computes the number of clusters in a graph, bridging the gap between the previous high time complexity for determining $k$ and the nearly-linear time algorithms for spectral clustering. In addition to its theoretical guarantees, our algorithm demonstrates strong empirical performance, with our experimental results validating its near-linear runtime in practice.

**Related Work.** There are many works that analyse the performance of spectral clustering on practical graph instances, e.g., (Kolev & Mehlhorn, 2016; Peng et al., 2017; Dey et al., 2019; Laenen & Sun, 2020; Mizutani, 2021; Macgregor & Sun, 2022). These analyses are typically based on the

---

[1]We say that a graph algorithm runs in nearly-linear time if it runs in $O(m \cdot \mathrm{poly} \log n)$ time, where $m$ and $n$ are the number of edges and vertices of the input graph. For simplicity, we use $\widetilde{O}(\cdot)$ to hide a poly-logarithmic factor of $n$.

existence of a large gap between the $(k+1)$th eigenvalue of a graph matrix and the $k$-way expansion of the underlying graph. Similar gap conditions are also widely used to analyse other clustering algorithms, e.g., (Zhu et al., 2013; Gharan & Trevisan, 2014; Czumaj et al., 2015; Peng, 2020; Laenen et al., 2023).

Our work bears some similarity to the generalised matrix rank estimation problem (Zhang et al., 2015; Ubaru et al., 2017) and the the spectral density estimation problem (SDE) (Lin et al., 2016; Jin et al., 2024). Our analysis further employs the techniques developed in the SDE literature (Braverman et al., 2022). However, unlike these problems, our problem needs to determine the *exact* value of $k$, assuming the presence of a cluster structure of the underlying graph.

## 2 PRELIMINARIES

### 2.1 NOTATION

Let $G = (V, E, w)$ be an undirected graph with $|V| = n$ vertices, $|E| = m$ edges, and weight function $w : V \times V \to \mathbb{R}_{\geq 0}$. For any edge $e = \{u, v\} \in E$, we write $w_G(u, v)$ or $w_G(e)$ to express the weight of $e$. For a vertex $u \in V$, we denote its *degree* by $\deg_G(u) \triangleq \sum_{v \in V} w_G(u, v)$, and the volume for any $S \subseteq V$ is defined as $\mathrm{vol}_G(S) \triangleq \sum_{u \in S} \deg_G(u)$. For any $S, T \subset V$, we define the *cut value* between $S$ and $T$ by $w_G(S, T) \triangleq \sum_{e \in E_G(S,T)} w_G(e)$, where $E_G(S, T)$ is the set of edges between $S$ and $T$. Moreover, for any $S \subset V$, the conductance of $S$ is defined as

$$\Phi_G(S) \triangleq \frac{w_G(S, V \setminus S)}{\min\{\mathrm{vol}_G(S), \mathrm{vol}_G(V \setminus S)\}}$$

if $S \neq \emptyset$, and $\Phi_G(S) = 1$ if $S = \emptyset$. For any integer $k \geq 2$, we call subsets of vertices $A_1, \ldots, A_k$ a $k$-way partition of $G$ if $\bigcup_{i=1}^k A_i = V$ and $A_i \cap A_j = \emptyset$ for different $i$ and $j$. We define the $k$-way *expansion* of $G$ by

$$\rho_G(k) \triangleq \min_{\text{partitions } A_1, \ldots, A_k} \max_{1 \leq i \leq k} \Phi_G(A_i).$$

Our analysis is based on matrix representations of graphs. For any graph $G$, let $D_G \in \mathbb{R}^{n \times n}$ be the diagonal matrix defined by $D_G(u, u) = \deg_G(u)$ for all $u \in V$. We denote by $A_G \in \mathbb{R}^{n \times n}$ the *adjacency matrix* of $G$, where $A_G(u, v) = w_G(u, v)$ for all $u, v \in V$. The *normalised adjacency matrix* of $G$ is defined by $N_G \triangleq D_G^{-1/2} A_G D_G^{-1/2}$. For any real and symmetric matrix $A$, we write the eigenvalues of $A$ as $\lambda_1(A) \geq \lambda_2(A) \geq \cdots \geq \lambda_n(A)$. We write the spectral norm of $A \in \mathbb{R}^{n \times n}$ by $\|A\|_2 \triangleq \max_{\substack{x:x \in \mathbb{R}^n \\ x \neq \mathbf{0}}} \|Ax\|/\|x\|$, and the Frobenius norm of $A \in \mathbb{R}^{n \times n}$ by $\|A\|_F \triangleq \sqrt{\sum_{1 \leq i,j \leq n} A_{i,j}^2} = \sqrt{\sum_{i=1}^n |\lambda_i(A)|^2}$. The following inequality builds the relationship between $1 - \lambda_k(N_G)$ and $\rho_G(k)$.

**Lemma 2** (Higher-order Cheeger inequality, (Lee et al., 2014)). *There is an absolute constant $C$ such that it holds for any graph $G$ and $k \geq 2$ that*

$$\frac{1 - \lambda_k(N_G)}{2} \leq \rho_G(k) \leq C \cdot k^3 \sqrt{1 - \lambda_k(N_G)}. \tag{1}$$

For any matrix $A \in \mathbb{R}^{n \times n}$ with the eigen-decomposition $Q\Lambda Q^*$ and function $f : \mathbb{R} \to \mathbb{R}$, let

$$f(A) \triangleq Qf(\Lambda)Q^*;$$

that is, we apply $f$ to the eigenvalues of $A$. For any real-valued functions $g$ and $h$ defined on $[-1, 1]$, we define

$$\langle g, h \rangle \triangleq \int_{-1}^1 g(x)h(x)\mathrm{d}x.$$

For any two functions $p$ and $q$ supported on $[-1, 1]$, the Wasserstein-1 distance between $p$ and $q$ is expressed by $W_1(p, q)$. By the dual formulation given by the Kantorovich-Rubinstein theorem (Kantorovich & Rubinshtein, 1957) we have

$$W_1(p, q) = \sup_{\substack{f : \mathbb{R} \to \mathbb{R} \\ |f(x) - f(y)| \leq |x-y| \forall x,y}} \left\{ \int_{-1}^1 f(x)(p(x) - q(x))\mathrm{d}x \right\}, \tag{2}$$

i.e., $p$ and $q$ are close in Wasserstein-1 distance if their difference has a small inner product with all 1-Lipschitz functions $f$.

## 2.2 Chebyshev Polynomials

Our work is based on the Chebyshev polynomials, the first type of which is defined recursively as follows: $T_0(x) = 1$, $T_1(x) = x$, and

$$T_k(x) = 2x \cdot T_{k-1}(x) - T_{k-2}(x)$$

for any $k \geq 2$. It's known that $\max_{x \in [-1,1]} |T_k(x)| \leq 1$ for any $k \geq 0$. The following lemma shows that the Chebyshev polynomials are orthogonal on $[-1, 1]$ under the weight function $w$, and the first $k$ Chebyshev polynomials form an orthogonal basis for the degree $k$ polynomials under this weight function:

**Lemma 3** ((Mudde, 2017)). *Let $w(x) \triangleq \frac{1}{\sqrt{1-x^2}}$. Then, the following holds: (1) $\langle T_0, w \cdot T_0 \rangle = \pi$, (2) $\langle T_k, w \cdot T_k \rangle = \pi/2$ for $k > 0$, and (3) $\langle T_i, w \cdot T_j \rangle = 0$ for any $i \neq j$.*

**Definition 4** (Chebyshev expansion). *For any function $f$ defined on $[-1, 1]$, the Chebyshev expansion of $f$ is defined as*

$$\sum_{k=0}^{\infty} \langle f, w \cdot \overline{T}_k \rangle \cdot \overline{T}_k, \tag{3}$$

*where $\overline{T}_k \triangleq T_k / \sqrt{\langle T_k, w \cdot T_k \rangle}$.*

The following alternative definition of Chebyshev polynomials will be used in our analysis.

**Definition 5** (Alternative definition of Chebyshev polynomial, (Mudde, 2017)). *A Chebyshev polynomial of degree $n$ is also defined as $T_n(x) = \cos(n \cdot \cos^{-1} x)$ for any $n \geq 0$.*

## 3 Algorithm

In this section we present and analyse the algorithm behind Theorem 1. Recall that graph $G$ has exactly $k$ clusters if (i) $G$ has $k$ disjoint subsets $A_1, \ldots, A_k$ of low conductance and hence a small value of $\rho_G(k)$, and (ii) any $(k+1)$-way partition of $G$ would include some $A \subset V$ of high conductance, which would be implied by a lower bound on $1 - \lambda_{k+1}(N_G)$ due to Lemma 2. To characterise the structure of clusters, we define

$$\Upsilon_G(k) \triangleq \frac{1 - \lambda_{k+1}(N_G)}{\rho_G(k)},$$

and a large value of $\Upsilon_G(k)$ shows that $G$ has exactly $k$ clusters. We prove that, when the input graph $G$ satisfies $\Upsilon_G(k) \geq C \cdot k$ for some $C \in \mathbb{R}^+$, the value of $k$ can be computed in nearly-linear time. Our result is as follows:

**Theorem 6** (Formal Statement of Theorem 1). *Let $G = (V, E)$ be an undirected graph with $n$ vertices and $m$ edges, and assume that $G$ satisfies that $\Upsilon_G(k) \geq C \cdot k$ for a universal constant $C \in \mathbb{R}^+$. Then, there is an algorithm that, given $G$ as input, runs in $\widetilde{O}(m)$ time and with probability at least $1 - o(1)$ returns the value of $k$.*

**Remark 1.** *Notice that $\Upsilon_G(k)$ is a well-studied quantity in spectral clustering (Kolev & Mehlhorn, 2016; Peng et al., 2017; Dey et al., 2019; Mizutani, 2021), and it's known that the performance of spectral clustering can be rigorously analysed for any graph $G$ with unbalanced clusters, as long as $\Upsilon_G(k) = \Omega(k)$ (Macgregor & Sun, 2022). We prove that, under the same condition, the value of $k$ can be determined in nearly-linear time.*

The design of our algorithm is based on three components: the first one is an efficient algorithm that constructs a sparse subgraph $H$ of $G$ such that both $G$ and $H$ have the same structure of clusters; this gives the normalised adjacency matrix $M$ of $H$. The second component is the procedure COUNTEIGENVALUES$(M, a, b)$, which counts the number of eigenvalues of $M$ in $[a, b]$. As the last one, our main algorithm invokes COUNTEIGENVALUES$(M, a, b)$ for different intervals and finds the right value of $k$.

### 3.1 Sparsification of $G$

Since an input graph $G$ could be a potentially dense graph and we only need to learn from the cluster structure of $G$, we first apply a sparsification algorithm of $G$ and obtain a sparse graph $H$, such that (i) $H$ has $\widetilde{O}(n)$ non-edges, and (ii) $G$ and $H$ have the same structure of clusters. We achieve this by constructing a cluster-preserving sparsifier of $G$.

**Definition 7** (Cluster-preserving sparsifier, (Sun & Zanetti, 2019))**.** *Let $G = (V, E, w_G)$ be any graph, and $\{A_i\}_{i=1}^{k}$ the k-way partition of $G$ corresponding to $\rho_G(k)$. We call a re-weighted sub-graph $H = (V, F \subset E, w_H)$ a cluster-preserving sparsifier of $G$ if (i) $\phi_H(A_i) = O(k \cdot \phi_G(A_i))$ for $1 \le i \le k$, and (ii) $1 - \lambda_{k+1}(N_H)$ and $1 - \lambda_{k+1}(N_G)$ differ by at most a constant factor.*

To construct a cluster-preserving sparsifier of $\widetilde{O}(n)$ edges, we apply the nearly-linear algorithm (Sun & Zanetti, 2019) described as follows: given any input graph $G = (V, E, w_G)$ with weight function $w_G$, the algorithm computes

$$p_u(v) \triangleq \min\left\{ C \cdot \frac{\log n}{1 - \lambda_{k+1}(N_G)} \cdot \frac{w_G(u,v)}{\deg_G(u)}, 1 \right\}$$

and

$$p_v(u) \triangleq \min\left\{ C \cdot \frac{\log n}{1 - \lambda_{k+1}(N_G)} \cdot \frac{w_G(v,u)}{\deg_G(v)}, 1 \right\}$$

for every edge $e = \{u, v\}$, where $C \in \mathbb{R}^+$ is some constant. Afterwards, the algorithm samples every edge $e = \{u, v\}$ with probability

$$p_e \triangleq p_u(v) + p_v(u) - p_u(v) \cdot p_v(u),$$

and sets the weight of every sampled $e = \{u, v\}$ in $H$ as $w_H(u, v) \triangleq w_G(u, v)/p_e$. This constructs the graph $H = (V, F, w_H)$. We use $M$ to denote the normalised adjacency matrix of $H$.

By our assumption $\Upsilon_G(k) \ge C \cdot k$ for a universal constant $C \in \mathbb{R}^+$ and the two properties of $H$ in Definition 7, the values of $\lambda_k(M)$ and $\lambda_{k+1}(M)$ differ by at least a constant. Without loss of generality we assume that $\lambda_k(M) \ge 2\beta \cdot \lambda_{k+1}(M)$ for $\beta > 2$. Moreover, it holds by construction that $\|M\|_2 \le 1$.

**Remark 2.** *Our chosen factor $2\beta$ is only used to simplify the presentation, and with the same time complexity our algorithm works as long as $\lambda_k(M)/\lambda_{k+1}(M) \ge \beta$ for any $\beta \in \mathbb{R}^+$ with $\beta > 1$.*

### 3.2 The CountEigenvalues Procedure

Next we study the problem of computing the number of eigenvalues of $M$ belonging to $[a, b]$, for some $0 < a, b \le 1$. Without loss of generality, we fix an arbitrary interval $[a, b]$ throughout the analysis, and define the spectral density of $M$ by

$$s(x) \triangleq \frac{1}{n} \sum_{i=1}^{n} \delta(x - \lambda_i), \tag{4}$$

where $\delta$ is a Dirac delta function. Notice that function $s$ gives every distinct eigenvalue the same probability mass of $1/n$, and the number of eigenvalues in $[a, b]$ equals to

$$\int_a^b n \cdot s(x)\mathrm{d}x.$$

We define a step function $h_{a,b}$ by

$$h_{a,b}(t) \triangleq \begin{cases} 1 & \text{if } t \in [a, b] \\ 0 & \text{otherwise,} \end{cases}$$

and this implies that

$$\int_a^b n \cdot s(x)\mathrm{d}x = \mathrm{tr}(h_{a,b}(M)).$$

Applying the Chebyshev expansion (Definition 4), we have that

$$\text{tr}(h_{a,b}(t)) = \sum_{i=0}^{\infty} \langle h_{a,b}, w \cdot \overline{T}_i \rangle \cdot \text{tr}\left(\overline{T}_i(t)\right). \tag{5}$$

We first prove that that the coefficients of the Chebyshev expansion above can be computed in $O(1)$ time.

**Lemma 8.** *It holds for any $i \geq 0$ that*

$$\langle h_{a,b}, w \cdot T_i \rangle = \begin{cases} \sin^{-1}(b) - \sin^{-1}(a) & \text{if } i = 0 \\ \frac{1}{i} \cdot \left(\sin(i \cos^{-1} a) - \sin(i \cos^{-1} b)\right) & \text{if } i > 0. \end{cases}$$

*Proof.* By definition, we have that

$$\langle h_{a,b}, w \cdot T_i \rangle = \int_{-1}^{1} \frac{h_{a,b}(x) \cdot T_i(x)}{\sqrt{1 - x^2}} \mathrm{d}x.$$

The proof is by case distinction.

*Case of $i = 0$:* By definition it holds that

$$\langle h_{a,b}, w \cdot T_0 \rangle = \int_{-1}^{1} \frac{h_{a,b}(x) \cdot T_0(x)}{\sqrt{1 - x^2}} \mathrm{d}x = \int_{a}^{b} \frac{1}{\sqrt{1 - x^2}} \mathrm{d}x = \sin^{-1}(b) - \sin^{-1}(a),$$

where the second line follows by the fact that $T_0(x) = 1$ and the definition of $h$.

*Case of $i > 0$:* By definition we have that

$$\langle h_{a,b}, w \cdot T_i \rangle = \int_{-1}^{1} \frac{h_{a,b}(x) \cdot T_i(x)}{\sqrt{1 - x^2}} \mathrm{d}x = \int_{a}^{b} \frac{T_i(x)}{\sqrt{1 - x^2}} \mathrm{d}x.$$

By Definition 5, we have $T_i(x) = \cos\left(i \cdot \cos^{-1} x\right)$. We set $x = \cos\theta$, and have that $\mathrm{d}x = -\sin\theta\mathrm{d}\theta$. Hence, substituting $x$ with $\cos\theta$ gives us that

$$\begin{aligned} \langle h_{a,b}, w \cdot T_i \rangle &= -\int_{\cos^{-1} a}^{\cos^{-1} b} \frac{\cos(i \cdot \cos^{-1}(\cos\theta))}{\sqrt{1 - \cos^2\theta}} \sin\theta\mathrm{d}\theta \\ &= -\int_{\cos^{-1} a}^{\cos^{-1} b} \cos(i \cdot \theta)\mathrm{d}\theta \\ &= -\int_{i \cdot \cos^{-1} a}^{i \cdot \cos^{-1} b} \frac{\cos x}{i}\mathrm{d}x \\ &= -\left[\frac{\sin x}{i}\right]_{i \cdot \cos^{-1} a}^{i \cdot \cos^{-1} b} \\ &= \frac{\sin(i \cos^{-1} a) - \sin(i \cos^{-1} b)}{i}, \end{aligned}$$

which proves the lemma. $\qquad\square$

Therefore, it suffices to study a fast approximation of $\text{tr}(T_i(M))$ and the number of leading terms $N$ needed in (5) to achieve a good approximation of $\int_a^b n \cdot s(x)\mathrm{d}x$. That is, we would like to know the order of $N$ for which

$$\int_a^b n \cdot s(x)\mathrm{d}x = \text{tr}(h_{a,b}(M)) = \sum_{i=0}^{\infty} \alpha_i \cdot \text{tr}(T_i(M)) \approx \sum_{i=0}^{N} \alpha_i \cdot \text{tr}(T_i(M)), \tag{6}$$

where

$$\alpha_i \triangleq \langle h_{a,b}, w \cdot \overline{T}_i \rangle.$$

**Approximating** $\text{tr}(T_i(M))$**.** For a general matrix $M$, computing $T_i(M)$ exactly requires matrix multiplication operations due to the recursive definition of $T_i(M)$, and a fast approximation of $\text{tr}(T_i(M))$ needs to avoid the explicit computation of $T_i(M)$. To achieve this, we employ the Hutchinson's estimator (Hutchinson, 1989), and its main idea is as follows: if one picks a random vector $x$ satisfying $\mathbb{E}\left[xx^\top\right] = \mathbb{I}$, then it holds that $\mathbb{E}\left[x^\top M x\right] = \text{tr}(M)$. To increase the accuracy of the estimator one can pick $\ell$ sub-Gaussian random vectors $x_1, x_2, \ldots, x_\ell$ and return

$$H_\ell(M) = \frac{1}{\ell} \sum_{i=1}^{\ell} (x_i)^\top M x_i = \frac{1}{\ell} \cdot \text{tr}\left(\text{X}^\top M \text{X}\right), \tag{7}$$

where $\text{X} = [x_1, x_2, \ldots, x_\ell]$ consists of $\ell$ independent copies of $x$. By applying the Hutchinson's estimator we estimate $\int_a^b n \cdot s(x)\mathrm{d}x$ by the quantity

$$\frac{1}{\ell} \sum_{j=1}^{\ell} \sum_{i=0}^{N} \alpha_i \left( x_j{}^\top T_i(M) x_j \right). \tag{8}$$

Notice that we need $O(N \cdot \ell)$ matrix-vector multiplications to calculate (8). Since $M$ has $\widetilde{O}(n)$ nonzero entries, the overall calculation takes $\widetilde{O}(N \cdot \ell \cdot n)$ time. The following lemma proves the approximation ratio of $H_\ell(T_k(M))$ with respect to $\ell$.

**Lemma 9.** *If we pick $\ell = O\left(\frac{1}{\epsilon^2} \cdot \log \frac{1}{\delta}\right)$ random vectors, in which every entry is sub-Gaussian, then for $k \geq 0$ it holds with probability at least $1 - \delta$ that*

$$\left|\text{tr}\left(T_k(M)\right) - H_\ell(T_k(M))\right| \leq \epsilon \cdot \sqrt{n}.$$

The following result is used in our analysis.

**Lemma 10** (Theorem A.1, Persson et al. (2022))**.** *Let $M \in \mathbb{R}^{n \times n}$ be symmetric, and $x \in \mathbb{R}^n$ a standard Gaussian vector. Then, for any $c \in (0, 1/2)$ and $C = -\frac{1}{c} - \frac{\log(1-2c)}{2c^2}$ it holds that*

$$\mathbb{P}\left(\left|x^\top M x - \text{tr}(M)\right| \geq \Delta\right) \leq 2 \cdot \exp\left(-\min\left\{\frac{\Delta^2}{4C\|M\|_F^2}, \frac{c\Delta}{2\|M\|_2}\right\}\right).$$

*Proof of Lemma 9.* Let $\bar{M} \in \mathbb{R}^{\ell n \times \ell n}$ be a block-diagonal matrix defined by

$$\bar{M} = \begin{pmatrix} T_k(M) & 0 & \cdots & 0 \\ 0 & T_k(M) & \cdots & 0 \\ \vdots & \vdots & \ddots & \vdots \\ 0 & 0 & \cdots & T_k(M) \end{pmatrix}.$$

Let $\text{X} \in \mathbb{R}^{n \times \ell}$ be the matrix of $\ell$ random vectors in (7), and $\text{x} = [x_1, x_2, \ldots, x_\ell] \in \mathbb{R}^{n\ell}$ be the vector representation of X. Then, it holds for every $k \geq 0$ that

$$\ell \cdot H_\ell(T_k(M)) = \text{tr}\left(\text{X}^\top T_k(M)\text{X}\right) = \text{x}^\top \bar{M}\text{x}. \tag{9}$$

By Lemma 10, we have

$$\mathbb{P}\left(\left|\text{x}^\top \bar{M}\text{x} - \text{tr}(\bar{M})\right| \geq \Delta\right) \leq 2 \cdot \exp\left(-\min\left\{\frac{\Delta^2}{4C\|\bar{M}\|_F^2}, \frac{c\Delta}{2\|\bar{M}\|_2}\right\}\right)$$

$$\leq 2 \cdot \exp\left(-\min\left\{\frac{\Delta^2}{4C\ell\|T_k(M)\|_F^2}, \frac{c\Delta}{2\|T_k(M)\|_2}\right\}\right)$$

$$\leq 2 \cdot \exp\left(-\min\left\{\frac{\Delta^2}{4C\ell n}, \frac{c\Delta}{2}\right\}\right), \tag{10}$$

where the second inequality follows by $\|\bar{M}\|_F^2 = \ell \cdot \|T_k(M)\|_F^2$ and $\|\bar{M}\|_2 = \|T_k(M)\|_2$, and the third one follows by $\|T_k(M)\|_2 \leq 1$ for every $k \geq 0$. We combine (9) with (10), and obtain that

$$\mathbb{P}\left(\left|\ell \cdot H_\ell(T_k(M)) - \ell \cdot \text{tr}(T_k(M))\right| \geq \Delta\right) \leq 2 \cdot \exp\left(-\min\left\{\frac{\Delta^2}{4C\ell n}, \frac{c\Delta}{2}\right\}\right),$$

which implies that

$$\mathbb{P}\Big(\big|H_\ell(T_k(M)) - \text{tr}(T_k(M))\big| \geq \frac{\Delta}{\ell}\Big) \leq 2 \cdot \exp\Big(-\min\Big\{\frac{\Delta^2}{4C\ell n}, \frac{c\Delta}{2}\Big\}\Big).$$

By setting $\Delta = \epsilon \cdot \ell \cdot \sqrt{n}$, we have

$$\mathbb{P}\Big(\big|H_\ell(T_k(M)) - \text{tr}(T_k(M))\big| \geq \epsilon \cdot \sqrt{n}\Big) \leq 2 \cdot \exp\Big(-\min\Big\{\frac{\ell \cdot \epsilon^2}{4C}, \frac{c\epsilon \cdot \ell \cdot \sqrt{n}}{2}\Big\}\Big) \leq 2 \cdot \exp\Big(-\frac{\ell \cdot \epsilon^2}{4C}\Big).$$

Hence, by setting $\ell = \frac{4C}{\epsilon^2} \cdot \log\frac{2}{\delta} = O\big(\frac{1}{\epsilon^2} \cdot \log\frac{1}{\delta}\big)$, we have

$$\mathbb{P}\Big(\big|H_\ell(T_k(M)) - \text{tr}(T_k(M))\big| \geq \epsilon \cdot \sqrt{n}\Big) \leq \delta. \qquad \square$$

**Upper bound of $N$.** We study the number of terms in (5) needed to achieve a good approximation. Notice it holds for every $\overline{T}_k$ in (5) that

$$\langle s, \overline{T}_k \rangle = \int_{-1}^{1} \frac{1}{n} \sum_{i=1}^{n} \delta(x - \lambda_i) \cdot \overline{T}_k(x) \mathrm{d}x = \frac{1}{n} \sum_{i=1}^{n} (\overline{T}_k(\lambda_i)) = \frac{1}{n} \text{tr}(\overline{T}_k(M)).$$

On the other hand, by the Hutchinson's estimator we implicitly obtain an function $q$ that satisfies $\langle q, \overline{T}_k \rangle = \frac{1}{n} H_\ell(\overline{T}_k(M))$. Hence, $W_1(s, q) \leq \epsilon$ implies that the algorithm returns the correct number of eigenvalues of $M$ in $[a, b]$. We prove that, by setting $N = \Theta(1/\epsilon)$, the statement shown in Lemma 9 implies that $W_1(s, q) \leq \epsilon$.

**Lemma 11.** *Let $A \in \mathbb{R}^{n \times n}$ be any symmetric matrix satisfying*

$$\frac{1}{n} \cdot |\text{tr}(T_k(A)) - H_\ell(T_k(A))| \leq \frac{1}{N \ln(eN)}.$$

*Then, it holds for $N = \Theta(1/\epsilon)$ that $W_1(s, q) \leq \epsilon$.*

To prove Lemma 11, we use the following two properties of a Lipschitz continuous function.

**Lemma 12** (Braverman et al. (2022)). *Let $f$ be a Lipschitz continuous function on $[-1, 1]$ with Lipschitz constant $\lambda > 0$. Then, for every $N \in 4\mathbb{N}^+$, there exists $N + 1$ constants $\hat{b}_N[0] > \ldots > \hat{b}_N[N] \geq 0$ such that the polynomial*

$$\bar{f}_N = \sum_{k=0}^{N} \frac{\hat{b}_N[k]}{\hat{b}_N[0]} \langle f, w \cdot \overline{T}_k \rangle \overline{T}_k$$

*satisfies that $\max_{x \in [-1,1]} |f(x) - \bar{f}_N(x)| \leq 18\lambda/N$.*

**Lemma 13** (Braverman et al. (2022)). *Let $f$ be a Lipschitz continuous function on $[-1, 1]$ with Lipschitz constant $\lambda > 0$. Then, it holds for any $k \geq 1$ that*

$$\big|\langle f, w \cdot \overline{T}_k \rangle\big| = \left|\int_{-1}^{1} f(x) \overline{T}_k(x) w(x) \mathrm{d}x\right| \leq 2\lambda/k.$$

*Proof of Lemma 11.* Since $s$ and $q$ are both supported on $[-1, 1]$, by (2) we have

$$W_1(s, q) = \sup_{\substack{f:\mathbb{R}\to\mathbb{R} \\ |f(x)-f(y)|\leq|x-y|\forall x,y}} \left\{\int_{-1}^{1} f(x)(s(x) - q(x)) \mathrm{d}x\right\}.$$

Let $f$ be an arbitrary 1-Lipschitz function, and $\big\{\hat{b}_N[k]\big\}_{k=0}^{N}$ and $\bar{f}_N$ be the coefficients and polynomial defined by Lemma 12 for function $f$. Using the triangle inequality we have

$$W_1(s, q) \leq \int_{-1}^{1} \big|f(x) - \bar{f}_N(x)\big| (s(x) - q(x)) \mathrm{d}x + \int_{-1}^{1} \bar{f}_N(s(x) - q(x)) \mathrm{d}x. \qquad (11)$$

Since $f$ is 1-Lipschitz function ($\lambda = 1$), by Lemma 12 we have

$$\int_{-1}^{1} \left| f(x) - \bar{f}_N(x) \right| (s(x) - q(x)) \mathrm{d}x \leq \frac{18}{N} \int_{-1}^{1} (s(x) - q(x)) \mathrm{d}x \leq \frac{36}{N}.$$

To bound the second term of (11) we use the Chebyshev series expansion of $\bar{f}_N$ and have that

$$\int_{-1}^{1} \bar{f}_N(x) w(x) \cdot \frac{s(x) - q(x)}{w(x)} \mathrm{d}x$$

$$= \int_{-1}^{1} \bar{f}_N(x) w(x) \cdot \sum_{k=0}^{\infty} \left\langle s - q, \overline{T}_k \right\rangle \overline{T}_k(x) \mathrm{d}x$$

$$= \int_{-1}^{1} \left( w(x) \sum_{k=0}^{N} \frac{\hat{b}_N[k]}{\hat{b}_N[0]} \left\langle f, w \cdot \overline{T}_k \right\rangle \overline{T}_k(x) \right) \left( \sum_{k=0}^{\infty} \left\langle s - q, \overline{T}_k \right\rangle \overline{T}_k(x) \right) \mathrm{d}x$$

$$\leq \sum_{k=1}^{N} \left| \left\langle f, w \cdot \overline{T}_k \right\rangle \right| \cdot \left| \left\langle \overline{T}_k, s \right\rangle - \left\langle \overline{T}_k, q \right\rangle \right|$$

$$\leq \sum_{k=1}^{N} \frac{2}{k} \cdot \left| \left\langle \overline{T}_k, s \right\rangle - \left\langle \overline{T}_k, q \right\rangle \right|, \tag{12}$$

where the first inequality follows by the orthogonality of the Chebyshev polynomials under the weight function $w$, the fact that $\left\langle \overline{T}_k, w \cdot \overline{T}_k \right\rangle = 1$ for all $k \in [N]$ as $0 \leq \hat{b}_N[k]/\hat{b}_N[0] \leq 1$ and

$$\left| \int_{-1}^{1} \overline{T}_k(s(x) - q(x)) \mathrm{d}x \right| = \left| \left\langle \bar{T}_k, s \right\rangle - \left\langle \bar{T}_k, q \right\rangle \right|$$

for each $k \in [N]$. The last inequality of (12) follows from Lemma 13. Combining these with (11) gives us that

$$W_1(s, q) \leq \frac{36}{N} + 2 \cdot \sum_{k=1}^{N} \frac{\left| \left\langle \overline{T}_k, s \right\rangle - \left\langle \overline{T}_k, q \right\rangle \right|}{k}$$

$$\leq \frac{36}{N} + 2 \cdot \sum_{k=1}^{N} \frac{1}{n} \cdot \frac{\left| \mathrm{tr}(T_k(M)) - H_\ell(T_k(M)) \right|}{k}$$

$$\leq \frac{36}{N} + \frac{2}{N \ln(\mathrm{e}N)} \cdot \sum_{k=1}^{N} 1/k$$

$$\leq \frac{36}{N} + \frac{2}{N \ln(\mathrm{e}N)} \cdot (\ln(\mathrm{e}N)) = \frac{38}{N},$$

where the third inequality follows by the condition of the lemma[2] and the fourth one follows by the fact that $H_n \leq 1 + \ln n$. By setting $N = 38/\epsilon = O(1/\epsilon)$, we have that $W_1(s, q) \leq \epsilon$. $\qquad\square$

**The COUNTEIGENVALUES Procedure.** Combining the Hutchinson's estimator and the Chebyshev expansion, our designed COUNTEIGENVALUES procedure is described in Algorithm 1. Notice that, as the Hutchinson's estimator computes $x^\top H_i(M)x$ for every $0 \leq i \leq N$, Algorithm 1 computes $x^\top H_i(M)x$ inductively for every $i$ and matrix multiplication operations are not needed.

**Lemma 14.** *Given any matrix $M \in \mathbb{R}^{n \times n}$ with $\|M\|_2 \leq 1$ and $\widetilde{O}(n)$ non-zero entries, parameters $0 \leq a, b \leq 1$, and $\epsilon \in (0, 1)$ as input, COUNTEIGENVALUES$(M, a, b)$ runs in $\widetilde{O}(n/\epsilon^3)$ time and outputs the number of $M$'s eigenvalues in $[a, b]$ with probability at least $1 - O(\epsilon/n)$.*

*Proof.* By setting $\delta = \epsilon/n$ in Lemma 9, we pick $\ell = O((1/\epsilon^2) \cdot \log(n/\epsilon))$ random vectors to construct a Huntchinson's estimator, and for every $k$ it holds with probability at least $1 - O(\epsilon/n)$ that

$$\left| \mathrm{tr}(T_k(M)) - H_\ell(T_k(M)) \right| \leq \epsilon \cdot \sqrt{n},$$

---

[2]Notice that the condition easily holds due to Lemma 9.

---

**Algorithm 1:** COUNTEIGENVALUES$(M, a, b, \epsilon)$

---

**Input:** Matrix $M \in \mathbb{R}^{n \times n}$, range of interval $[a, b]$, and parameters $\epsilon$
**Output:** the number of eigenvalues of $M$ in $[a, b]$
1:  **for** $i = 1$ **to** $\ell = O\left((1/\epsilon^2) \cdot \log(\epsilon/n)\right)$ **do**
2:     pick $x \in \mathbb{R}^n$ with i.i.d. $\{-1, 1\}$ entries
3:     $x_0 \leftarrow x$, $\mathbb{T}[i, 0] \leftarrow x_0^\top x_0$
4:     $x_1 \leftarrow A \cdot x_0$, $\mathbb{T}[i, 1] \leftarrow x_0^\top x_1$
5:     **for** $k = 2$ **to** $N$ **do**
6:        $x_k \leftarrow 2 \cdot M \cdot x_{k-1} - x_{k-2}$
7:        $\mathbb{T}[i, k] \leftarrow x_0^\top x_k$
8:     **end for**
9: **end for**
10: **return** $\frac{1}{\ell} \sum_{i=1}^{\ell} \sum_{j=1}^{N} \alpha_j \cdot \mathbb{T}[i, j]$         // Here $\alpha_k$s are computed from Lemma 8.

---

implying that the precondition of Lemma 11 holds for every $k$. Hence, the claimed success probability follows by taking the union bound of $N$ applications of the Huntchinson's estimator. The total running time follows by our choice of $N = \Theta(1/\epsilon)$ and $\ell = O((1/\epsilon^2) \cdot \log(n/\epsilon))$.  □

### 3.3 THE MAIN ALGORITHM AND THE PROOF OF THEOREM 6

Our main algorithm is based on repeated executions of COUNTEIGENVALUES with different parameters, and consists of the two phases:

- the algorithm invokes COUNTEIGENVALUES $\left(M, 1 - (\beta/2)^i/n^2, 1\right)$ for $i = 1, 2, 3, \ldots$, until the output of COUNTEIGENVALUES $\left(M, 1 - (\beta/2)^{i'}/n^2, 1\right)$ for some $i' \in \mathbb{N}$ is at least 2.

- the algorithm continues to invoke COUNTEIGENVALUES $\left(M, 1 - (\beta/2)^i/n^2, 1\right)$ from $i = i'$, and terminates when any two executive executions return the same value. The algorithm outputs this value as the number of clusters in $G$.

Now we analyse the correctness of the algorithm. We apply the first phase to find the smallest $i$ such that $\lambda_2 \in \left[1 - (\beta/2)^i/n^2, 1\right]$ and this approximate value of $\lambda_2$ is needed for the algorithm, since otherwise COUNTEIGENVALUES could simply return 1 for the first two executions while only counting $\lambda_1(M)$. Then, in the second phase the algorithm returns the correct value of $k$ by our assumption that $\lambda_k(M) \geq 2\beta \cdot \lambda_{k+1}(M)$ for $\beta > 2$.

Secondly, we analyse the time complexity of our algorithm. Since constructing a cluster-preserving sparsifier takes $\widetilde{O}(m)$ time (Sun & Zanetti, 2019) and our main algorithm runs $O(\log n)$ times of the COUNTEIGENVALUES procedure with time complexity $\widetilde{O}(n/\epsilon^3)$, the total running time of the algorithm is $\widetilde{O}(m + n/\epsilon^3)$.

Finally, we analyse the success probability of the algorithm. Since we run COUNTEIGENVALUES $O(\log n)$ times and every execution returns the correct value with probability at least $1 - O(\log^c n/n)$ for some constant $c$, taking the union bound proves the success probability of our main algorithm. This proves Theorem 6.

## 4   EXPERIMENTS

This section evaluates the performance of our designed algorithm. Since our algorithm is the first nearly-linear time algorithm for the problem, the primary goals of our experiments are to demonstrate the nearly-linear running time of the algorithm in practice, and its effectiveness in determining the value of $k$. All of our experiments were performed on a Lenovo Yoga 2 Pro with an Intel(R) Core(TM) i7-4510U CPU @ 2.00GHz processor and 32GB of RAM. Our reported running times are averaged over 5 runs of the algorithm.

We first evaluate the performance of our algorithm on the Stochastic Block Model (SBM) with parameter $n, k, p, q$ (Abbe, 2018). A random graph $G$ generated from the SBM has $k$ clusters, each of which contains $n$ vertices. Moreover, every pair of vertices within the same cluster is connected with probability $p$, and every pair of vertices belonging to different clusters is connected with probability $q$. All of our tested graphs are randomly generated from the SBM with the STAG library (Macgregor & Sun, 2024). We evaluate the performance of our algorithm on three sets of input instances:

1. we set $k = 4$, $p = 0.6$, and $q = 0.1$, and run the algorithm with respect to $n$ between $2,000$ and $5,000$; this setup ensures that the total number of edges in $G$ is approximately linear in $n$. Figure 1a reports the running time of the algorithm with respect to the total number of edges in $G$;

2. we set $n = 2,000$, $k = 4$, $p = 0.6$, and increase the values of $q$. Notice that, as $q$ increases, the cluster structure of the input graph becomes less significant, making it more challenging for the algorithm to accurately determine the value of $k$. Figure 1b reports the running time of the algorithm with respect to the total number of edges in $G$;

3. we set $n = 500 \cdot k$, $p = 0.6$, and $q = 0.08$, and increase the value of $k$ from 2 to 8.

For all the tested instances our algorithm correctly determines $k$, and Figure 1 further demonstrates its nearly-linear running time in practice.

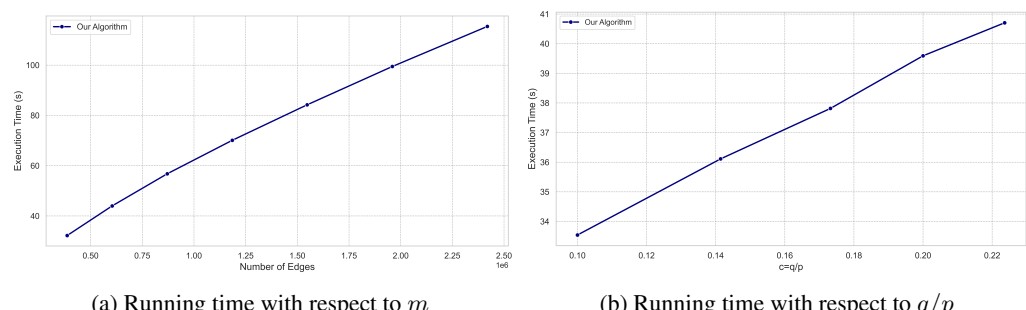

(a) Running time with respect to $m$        (b) Running time with respect to $q/p$

Figure 1: The running time of our algorithm with respect to (a) the total number of edges, and (b) the value of $q$ for a fixed $p$.

Secondly, we apply the `scikit-learn` library to generate 500 data points in $\mathbb{R}^2$ from classical clustering datasets. The data is created using the `make_circles`, `make_moons`, and `make_blobs` methods, each configured with a noise parameter of 0.05. The `make_circles` method is set with a factor of 0.5, controlling the distance between the inner and outer circles, and the `make_blobs` method generates two clusters with the standard deviation of 0.5 for each cluster. Every constructed data set gives us a graph with 500 vertices and approximately 70,000 edges, with the weights of the edges determined by the Euclidean distances between the data points. Our algorithm correctly identifies $k$ for every graph instance with the average running time of about 9 seconds.

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
