# OpenReview forum: "Finding the Number of Clusters in a Graph: a Nearly-Linear Time Algorithm"
_ICLR.cc/2025/Conference — Submitted to ICLR 2025_

### Official Review · Reviewer_3WEb · 2024-10-27

**Soundness:** 2
**Presentation:** 3
**Contribution:** 3
**Rating:** 5
**Confidence:** 3

**Summary:**

This paper introduces a nearly linear time algorithm designed to determine the number of clusters in a graph using the eigengap heuristic. The primary contribution is an algorithm that counts the eigenvalues of the normalized adjacency matrix within a specified interval [a,b]. The author then leverages graph sparsification and this eigenvalue-counting algorithm to estimate the number of clusters $k$.

**Strengths:**

The paper makes a valuable contribution to the scalability of spectral clustering for large graphs. Combined with [Peng et al., 2017], it demonstrates that both the eigen-gap heuristic to infer $k$ and spectral clustering to infer the clusters can be executed in nearly-linear time.

**Weaknesses:**

* The datasets used in the experimental evaluation are limited in size. Testing on larger graphs is necessary to conclude the algorithm's empirical runtime, especially as the paper emphasizes scalability as a key benefit. For example, testing on graphs with 10^4, 10^5, 10^6, and more vertices (or edges) would show how runtime empirically scales with graph size.

**Questions:**

* Remark 2: This section is somewhat unclear. In line 191, $\beta$ appears with a factor of 2 and requires $\beta > 2$, while line 194 does not include the factor of 2 and suggests $\beta > 1$. This may be a typo, as line 194 may need to specify $\beta > 2$ as well. Additionally, it seems this assumption primarily applies to the interval sizes in step (iii) of the algorithm. For clarity, it might be preferable to assume something like $\lambda_k(M) \ge (2+\epsilon) \lambda_{k+1}(M)$ for some $\epsilon > 0$ and proceed from there.

* It seems the authors' proposed sparsification step requires prior knowledge of $k$, given that the definitions of $p_{u}(v)$​ and $p_v(u)$​ on lines 177 and 180 rely on the quantity $ \lambda_{k+1}(N_{G}) $. Could the authors clarify if the method can avoid this dependency on $k$ or if alternative sparsification approaches, such as spectral sparsification by Spielman, Srivastava, or Teng, might also be suitable?

* Line 215: what is $h_{ab}(M)$? If $h_{ab}$ is applied entrywise, I am not sure to understand where the equality comes from.

* Line 321: What is the justification for $\|T_k(M)\|_2 \le 1$? Additionally, the third inequality in this line also assumes $\|T_k(M)\|_F^2 \le n$, which is not explicitly shown. Further explanation would be helpful.

* Line 494: If $p$ and $q$ are fixed while $n$ increases, should the number of edges grow at a rate of $n^2$ instead?


Minor Points:

Line 38: “graph with n”: missing the word “vertices.”

Line 42: “nearly-liner” should be corrected to “nearly-linear.”

Line 303: The symbol should likely be $\bar{M}$ rather than $\bar{A}$.

Line 464: The sentence in this line lacks clarity and may need rephrasing for coherence.

---

> ### Author Response · Authors · 2024-11-20
>
> We thank the reviewer for their careful reading and evaluation on our work. Here we respond to the weakness and questions raised from the report.
>
> > **Weakness:** The datasets used in the experimental evaluation are limited in size.
>
> **Response:** As the reviewer correctly pointed out that the purpose of our experiment is to conclude   the algorithm's empirical runtime, we believe that our reported Figure 1(a) suffices to  demonstrate that the runtime of our algorithm increases linearly in the number of edges of $G$, i.e., the visualised curve is  a linear function. However, if the reviewer thinks that experiments on larger graphs could be more helpful, we're happy to add experimental results on larger graphs in the next version of our paper.
>
> > **Question 1:** Remark 2: This section is somewhat unclear. In line 191,  $\beta$ appears with a factor of 2 and requires $\beta>2$, while line 194 does not include the factor of 2 and suggests $\beta>1$. This may be a typo, as line 194 may need to specify $\beta>2$
>  as well. Additionally, it seems this assumption primarily applies to the interval sizes in step (iii) of the algorithm. For clarity, it might be preferable to assume something like $\lambda_k(M)\geq (2+\epsilon)\lambda_{k+1}(M)$ for some $\epsilon>0$  and proceed from there.
>
> **Response:** The formulation of Remark 2 and the discussion on Line 194 is not a typo. In Remark 2, we assume that $\lambda_k(M)\geq 2\beta\cdot \lambda_{k+1}(M)$ for $\beta>2$; here we introduce the parameter $\beta$ and constant $2$ in order to simplify the analysis in Section 3.3. However, Line 194 states that our result actually holds as long as $\lambda_k(M)/\lambda_{k+1}(M)$ is lower bounded by some constant strictly greater than 1. We will rewrite these two sentences to make it clearer.
>
> > **Question 2:** It seems the authors' proposed sparsification step requires prior knowledge of $k$. Could the authors clarify if the method can avoid this dependency on $k$  or if alternative sparsification approaches, such as spectral sparsification by Spielman, Srivastava, or Teng, might also be suitable?
>
> **Response:** First of all, the algorithms by Spielman and Teng, as well as Spielman and Srivastava cannot be directly applied in our setting since, by the higher-order Cheeger inequality, we need to preserve the $(k+1)$-th smallest eigenvalue of the *normalised* Laplacian matrices, which cannot be directly implied by the above-mentioned algorithms. In addition, there is no efficient implementation of the both algorithms, and that's why we chose to use the Sun-Zanetti algorithm instead.
>
> Secondly, we highlight that our algorithm doesn't need prior knowledge of $k$. By the definition of $p_u(v)$,
>  a good approximation of $C\cdot \frac{\log n}{1-\lambda_{k+1}(N_G)}$ suffices for our purpose. Since $1-\lambda_{k+1}(N_G)=\Theta(1)$ when $G$ has $k$ clusters, we can treat $C\cdot \frac{\log n}{1-\lambda_{k+1}(N_G)}$ as $\mathrm{poly}\log(n)$, and this results in  the total number of sampled edges as $\widetilde{O}(n)$. It's important to notice that, with such approximation on the sampling probability, our work returns the *exact* value $k$; such exact value of $k$ is needed in spectral clustering.
>
> > **Question 3:** Line 215: what is $h_{ab}(M)$? If $h_{ab}$  is applied entrywise, I am not sure to understand where the equality comes from.
>
> **Response:** We define function $h_{a,b}: \mathbb{R} \rightarrow \mathbb{R}$ on Lines 211 -- 213, and Lines 095 -- 096 shows that how to generalise any function $f:\mathbb{R} \rightarrow \mathbb{R}$ to $f: \mathbb{R}^{n\times n} \rightarrow \mathbb{R}$. Here $f_{ab}$ applies to the eigenvalues of $M$.
>
> > **Question 4:** Line 321: What is the justification for $|T_k(M)|_2\leq 1$? Additionally, the third inequality in this line also assumes $|T_k(M)|_F^2\leq n$, which is not explicitly shown. Further explanation would be helpful.
>
> **Response:**  We first present the proof of $||T_k(M)||_2 \le 1$. Let the eigen-decomposition of $M$ be $Q \Lambda Q^*$, and we have by definition that  $T_k(M)= Q T_k(\Lambda) Q^{*}$. Since  $T_k(x)= \cos(n\cdot\cos^{-1} x)$, we have for any eigenvalue $\lambda_i$ of $M$  that $T_k(\lambda_i)\in[-1, 1]$  for any $k\geq 0$. This proves that $||T_k(M)||_2 \le 1$. The fact of $|| T_k(M)||_F^2\leq n$ follows $||T_k(M)||_2 \le 1$ and the definition of $||.||_F$. We will add more explanation on this in the next version of our submission.
>
> > **Question 5:** Line 494: If $p$ and $q$ are fixed while $n$ increases, should the number of edges grow at a rate of $n^2$ instead?
>
> **Response:** Thanks a lot for pointing out this. You're right that the number of edges grow at a rate of $n^2$ here, and we'll correct this in the next version of the paper. This change doesn't affect our claimed nearly-linear runtime time observed in practice.

---

> > ### Comment · Reviewer_3WEb · 2024-11-26
> >
> > Dear authors,
> >
> > Thank you for your detailed rebuttal and for the time you have dedicated to addressing the reviewers' concerns. While I appreciate your clarifications, I believe the paper's exposition could still benefit from further refinement—particularly in the technical details, such as clarifying the necessity (or lack thereof) of knowing $\lambda_{k+1}$. Given these issues, I feel a significant revision is still required, and therefore, I prefer to maintain my current score.

---

### Official Review · Reviewer_f2cJ · 2024-11-01

**Soundness:** 1
**Presentation:** 3
**Contribution:** 2
**Rating:** 3
**Confidence:** 5

**Summary:**

Given a graph $G(V,E)$ with $|V|=n$ and $|E|=m$ and $k$-many well-defined clusters, the paper aims to determine $k$ efficiently. The authors work under the assumption that the graph has a large eigengap, that is, the $k$-th eigenvalue of the normalized adjacency matrix $N_G$ is much larger than the $k+1$-th eigenvalue. In this setting, the authors provide an algorithm that guesses $k$ correctly with probability $0.9$  in $\mathcal{O}(m \cdot {\sf poly} \log (n) )$   (or $\tilde{\mathcal{O}}(n)$ ) time.

Below, I describe the steps of the algorithm and point out inconsistencies in the algorithm/proof arguments that I have observed.

---



**Step 1 (Section 3.1):** First, they use a "cluster preserving sparsifier" from existing work (Sun and Zanetti, 2019) that can sparsify the graph $G$ to $H$ with $\tilde{\mathcal{O}}(n)$ edges while ensuring the aforementioned eigengap is maintained.


> **First inconsistency:**  The sparsifier method itself needs the knowledge of $k$ (which the overall algorithm is trying to determine).

The sparsifier decides two values $p_u(v)$ and $p_v(u)$ for each edge $e(u,v)$. Then each edge is sampled with probability $p_u(v)+p_v(u)-p_u(v)*p_v(u)$. However, as described in lines *176* to *182* of the paper, $p_u(v)$ requires the knowledge of $\lambda_{k+1}(N_G)$, that is, the $k+1$-th eigenvalue of $N_G$ (where $k$ is the underlying number of clusters that the authors are trying to determine). This makes for a *circular* argument. One needs to know $k$ to obtain $\lambda_{k+1}(N_G)$, and the paper's goal is to determine $k$ itself.
This implies that either the algorithm is incorrect (if Sun and Zanetti themselves need the knowledge of $k$) or that Sun and Zanetti's paper can determine $k$ in linear time already, which makes this paper redundant.

----


**Step 2 (Section 3.2):** Then, the authors use recent analysis from [1] in a straightforward manner (with some modifications) to design Algorithm 1 of the paper, which given a matrix $M$, and two reals $a$ and $b$, counts the number of eigenvalues in the range $[a,b]$ in $\tilde{\mathcal{O}}(n)$ time.


----



**Step 3 (Section 3.3):** They want to count the number of eigenvalues of $N_H$ in different ranges using the previous algorithm, and they want to exploit the fact that if there is a large gap between $k$ and the $k+1$-th eigenvalue, the counting algorithm will return the same output ($k$) around this point even when the search range is expanded.

To do this, they search in the range $[1-(\beta)^i/n^2,1]$ for increasing $i$, starting from $i=1$. They first keep increasing $i$ until the returned count is greater than $1$ (to ensure they at least get the second eigenvalue in their range) and then keep increasing $i$ until they get the same count for some $[1-(\beta)^{i'}/n^2,1]$ and $[1-(\beta)^{i'+1}/n^2,1]$ (that is, from the time their search range found the second eigenvalue and up until this point every increase in $i$ results in more eigenvalue being found in the range), which indicates they have encountered the assumed eigengap. At this point, they stop their algorithm and return the number of eigenvalues counted in this range as the number of clusters $k$.

> **Second inconsistency (correctness):**  The authors do not prove that this algorithm cannot terminate for some $k'<k$ value.

The paper assumes that $\lambda_{k+1}(N_G)-\lambda_{k}(N_G)$ is large, and therefore $\lambda_{k+1}(N_H)-\lambda_{k}(N_H)$ is also large. Therefore, the counting algorithm would return the same count in this neighborhood. However, the paper does not argue why $\lambda_{j+1}(N_H)-\lambda_{j}(N_H)$  is sufficiently small for all $2\le j<k$ (so that count keeps increasing for every increasing $i$ until the range reaches the eigengap). Without such a guarantee, the correctness of the algorithm is not complete.

p.s. Here I note that if all the clusters are of the same size and are somewhat uniform (as in the SBM experiments by the author), then it is reasonable to believe that $ \lambda_{j+1}(N_H)-\lambda_{j}(N_H)$ is very small for $2\le j<k$. However this does not seem to hold in the general case.


---

[1] Vladimir Braverman, Aditya Krishnan, and Christopher Musco. Sublinear time spectral density estimation. In 54th Annual ACM Symposium on Theory of Computing (STOC’22), pp. 1144–1157, 2022.

**Strengths:**

Fast algorithms for unsupervised inference are always a welcome addition to the literature. The authors have also implemented the algorithms and showed some experimental results, which is nice. Given the mathematical inconsistencies I have observed, I cannot further comment on the paper's strengths. Overall it seems that the paper needs significant revisions before it can be published.

**Weaknesses:**

The primary weakness is the lack of clarity on the correctness of the algorithm that I have mentioned in the summary. To re-state:

1) The authors use a sparsifier from the work by Sun and Zanetti, 2019 which itself requires the value of the $k+1$-th eigenvalue of the original graph. This makes the current algorithm incorrect as the sparsifier itself needs the knowledge of $k$ (to obtain $k+1$-th eigenvalue.

2) The authors also do not provide a complete proof of the correctness in the last step of the algorithm (described in detail in the Summary).

**Questions:**

Could the authors comment on the inconsistencies in the algorithm/proof I mentioned in the Summary? I look forward to reading the author's explanations.

---

> ### Author Response · Authors · 2024-11-20
>
> We thank the reviewer for their careful reading and evaluation on our work. Here we respond to the concerns raised from the report.
>
> > **First inconsistency:** The sparsifier method itself needs the knowledge of $k$ (which the overall algorithm is trying to determine).
>
> We highlight that our algorithm doesn't need prior knowledge of $k$ due to the following reasons. By the definition of $
> p_u(v) $,
>  a good approximation of $C\cdot \frac{\log n}{1-\lambda_{k+1}(N_G)}$ suffices for our purpose. Since $1-\lambda_{k+1}(N_G)=\Theta(1)$ when $G$ has $k$ clusters, we can treat $C\cdot \frac{\log n}{1-\lambda_{k+1}(N_G)}$ as  $\mathrm{poly}\log(n)$ is sufficient, and this results in  the total number of sampled edges as $\widetilde{O}(n)$. It's important to notice that, with this approximation on the sampling probability, our work returns the *exact* value $k$; such exact value of $k$ is needed in spectral clustering.
>
> We further highlight that this is *not* a circular argument, and our work applies *reasonable approximation* of the sampling probability to achieve the *exact* value of $k$.
>
> > **Second inconsistency (correctness):** The authors do not prove that this algorithm cannot terminate for some $k'<k$ value.
>
> Thank you  for pointing out this technical question. To explain why there is no large gap between $\lambda_{j+1}(N_G)$ and $\lambda_j(N_G)$, notice that by the higher-order Cheeger inequality we have for any $2\leq j\leq k$ that
>     $
>      \rho^2(j)/j^6 \lesssim 1-\lambda_j(N_G) \lesssim \rho(j)$.
> On one side, if  $1-\lambda_j(N_G)$ is close to $\rho(j)$, then a large gap between $\lambda_{j+1}(N_G)$ and $\lambda_j(N_G)$
> implies a lower bound of $\Upsilon_G(j)$ and therefore $G$ has $j$ clusters; in this case our algorithm returns the correct number of clusters as output. On the other side, if $1-\lambda_j(N_G)$ is close to $\rho^2(j)$, then   Kwok et al. (2013) implies that $1- \lambda_j(N_G)\approx 1-\lambda_{j+1}(N_G)$. We will make this more formal in the next version of our paper.

---

> > ### Author Response · Authors · 2024-11-26
> >
> > Dear Reviewer,
> >
> > We are wondering if you had time to look at our response to the two inconsistency mentioned in your report. With our response, we believe that our main result still holds. However, if you think further clarification is needed, we can provide more details. Thank you!

---

> > > ### Comment · Reviewer_f2cJ · 2024-11-27
> > >
> > > Dear authors,
> > >
> > > Sorry for my delayed response. I tried to spend some time going through the papers you have cited to understand your claims better. After spending several hours going through the literature, I feel the paper needs a major revision before it can be considered for publication.
> > >
> > >
> > > 1) The authors say that $1-\lambda_{k+1}(G)=\Theta(1)$. Why is this true? Could the authors elaborate?
> > >
> > > 2) The proof of the convergence of the final iterative process should also be made formal. The arguments provided by the authors are somewhat handwavey, and I cannot validate their correctness.
> > >
> > > 3) I would like to ask why the authors chose to write their Laplacian as $D^{-1/2}AD^{-1/2}$ and then use $1-\lambda_{k}$ instead of defining the canonical way of $I-D^{-1/2}AD^{-1/2}$ and then directly using the eigenvalues? Of course, the two are identical, but most of the cited papers (such as the paper that describes the sparsification lemma) use the latter representation, which makes this paper harder to interpret in terms of existing results.

---

> > > > ### Author Response · Authors · 2024-11-27
> > > >
> > > > Thank you for the response. Here are our answers to your further questions.
> > > >
> > > > > The authors say that $1-\lambda_{k+1}(G)=\Theta(1)$. Why is this true? Could the authors elaborate?
> > > >
> > > > When $G$ has $k$ clusters, any $(k+1)$-way partition of $G$ would have to partition one cluster into two, and therefore $\rho(k+1)=\Omega(1)$. By the higher-order Cheever inequality, we have that
> > > > $$
> > > > 1-\lambda_{k+1} \leq 2\rho(k+1) =\Omega(1).
> > > > $$
> > > > Combining this with the trivial bound of $1-\lambda_{k+1} =O(1)$ proves that $1-\lambda_{k+1}(G)=\Theta(1)$.
> > > >
> > > > >The proof of the convergence of the final iterative process should also be made formal. The arguments provided by the authors are somewhat handwavey, and I cannot validate their correctness.
> > > >
> > > > Thank you for checking our answer and the reference very carefully. We will add more formal discussion in the next version of the paper. However, we feel that our response does answer your question.
> > > >
> > > > Could you be more specific about the places about which you cannot validate their correctness, or point out the specific place for which you think our analysis is incorrect? Otherwise, it would be impossible for us to answer your question.
> > > >
> > > > >I would like to ask why the authors chose to write their Laplacian as $D^{-1/2}AD^{-1/2}$  and then use $1-\lambda_i$ instead of defining the canonical way of $I-D^{-1/2}AD^{-1/2}$ and then directly using the eigenvalues? Of course, the two are identical, but most of the cited papers (such as the paper that describes the sparsification lemma) use the latter representation, which makes this paper harder to interpret in terms of existing results.
> > > >
> > > > This is a very good question, and we choose to use normalised adjacency matrix for technical reasons indeed. Notice that, if we consider normalized Laplacian matrix, then the first smallest eigenvalues could be as small as $O(1/n)$. As a result, in Lemma 11 we need to set $\epsilon$ to be $O(1/n)$, and then the running time of our algorithm won't be nearly-linear anymore. However, when using the normalised adjacency matrix, the top $k$ eigenvalues are close to $1$, and setting $\epsilon$ to be $O(1/\mathrm{poly}(\log n))$ suffices for our purpose.
> > > >
> > > > We remark that both of adjacency matrices and Laplacian matrices are widely used in spectral clustering literature. While some of our cited papers use Laplacian matrix representation, many papers use the adjacency matrix representation. One such example is the most classical paper on spectral clustering by Andrew Ng et al. (2001).

---

> ### Comment · Reviewer_f2cJ · 2024-11-28
> **Continued mistakes in the proof argument**
>
> The author's argument on $1-\lambda_{k+1}=\Omega(1)$ is incorrect.
>
> The authors write:
>
> 1) $\rho(k+1)=\Omega(1)$
> 2) $(1-\lambda_{k+1})$ **<=** $2 \rho(k+1) $
> 3) They claim 1 and 2 implies $1-\lambda_{k+1} = \Omega(1)$.
>
> This is incorrect. In order to show that $1-\lambda_{k+1} = \Omega(1)$, the authors need to show that $(1-\lambda_{k+1})$ is **greater than an $\Omega(1)$ quantity**.
>
> In general, if you have $y=\Omega(1)$ and $x \le 2y$ this does not imply $x= \Omega(1)$. For example, if $y=1$ and $x=1/n$ then the inequality $x \le 2y$ holds but $x=1/n$ which is not $\Omega(1)$.

---

> > ### Author Response · Authors · 2024-11-28
> >
> > Dear Reviewer,
> >
> > Sorry for the mistake we made in our last response to your question. Here is our new proof.
> >
> > By the higher-order Cheeger inequality, we have that
> > $$
> > \rho(k+1)\leq O((k+1)^3)\sqrt{ 1-\lambda_{k+1}  }.
> > $$
> > Since $\rho(k+1)=\Omega(1)$ and $k=O(\mathrm{poly}\log n)$ for most cases, we have that $1-\lambda_{k+1} = \Omega(1/\mathrm{poly}\log (n))$. With this, we can always treat $1-\lambda_{k+1}$ as $1/\mathrm{poly}\log (n)$ and the total sampled edges from the sparsification step remains $\tilde{O}(n)$.
> >
> > We do acknowledge that this updated analysis requires the assumption of $k=O(\mathrm{poly}\log n)$. While it's a reasonable assumption, we should have stated it in our original submission. Please let us know if any further clarification is needed. Thank you very much.

---

> ### Comment · Reviewer_f2cJ · 2024-11-28
>
> The proof is still incorrect. The authors claim $\rho(k+1)=\Omega(1)$. I do not see why this trivially holds. The primary assumption of the paper is that $\frac{1-\lambda_{k+1}}{\rho(k)} \ge C \cdot k$.
> This implies $(1-\lambda_{k+1}) \ge C \cdot k \cdot \rho(k)$.
>
> Furthermore, we only know that $\rho(k+1) \ge 2 (1-\lambda_{k+1})$.
> Then all we can conclude from here is that $\rho(k+1) \ge 2 (1-\lambda_{k+1}) \ge 2C \cdot k \cdot \rho(k) \implies \rho(k+1) \ge 2 \cdot C \cdot \rho(k)$ (this is also what the authors write in Lines 138-140).
>
> Note that $\rho(k)$ can be arbitrarily small (such as $n^{-\epsilon_1}$ for some constant $0<\epsilon_1<1$) and then the only guarantee you get is that $\rho(k+1)$ is greater than some $o(1)$ quantity. In fact, the lower bounding quantity can be as small as $n^{-\epsilon_2}$ for another $0<\epsilon_2<1$ as $k$ is at most ${\sf poly} \log n$. This **does not** guarantee that $\rho(k+1)= \Omega(1)$ (as the authors assume).
>
>
> ---
> ---
>
> Additionally, I have concerns about the significance of the results in light of the modified assumption of $k=\mathcal{O}({\sf poly} \log n)$. First of all, this now only considers an exponentially smaller set of $k$ values than the original paper (so theoretically, this is a significantly weaker claim). Furthermore, in this case, $\tilde{\mathcal{O}}(m)$ time complexity is confounding because really it can subsume the dependence on $k$. Still, these major concerns are superseded for now by the aforementioned mistake in the proof.
>
> ---
>
> In light of the continued mistakes in the proof argument, both in the original paper and in the responses, I believe a significant rewriting is needed, followed by a thorough review of the paper again. Therefore, I recommend the paper's rejection.

---

### Official Review · Reviewer_ySsE · 2024-11-01

**Soundness:** 4
**Presentation:** 4
**Contribution:** 3
**Rating:** 6
**Confidence:** 4

**Summary:**

Given an undirected graph that has a significant eigen gap between k-th and (k+1)-th largest eigenvalues, this paper gives a near-linear (in the number of edges) time algorithm for identifying k.

The algorithm is built on several known results, but the combination of these is natural and elegant. The first step is to reduce the problem to counting the number of eigenvalues in a given range [a, b]. For this task, this paper makes use of Chebyshev’s polynomial and expansion, to reduce this to a trace estimation problem. This problem is then solved by employing Hutchinson’s estimation, and this paper gives a new concentration bound of this estimation (specifically for their trace approximation problem).

Experiments are conducted to validate the performance of the proposed algorithm, on  synthesized datasets by stochastic block model and some other synthesized datasets generated by scikit-learn. The real world running time is reported.

**Strengths:**

- The result is clean, and the eigen gap assumption seems to be natural (and was also justified in previous works)

- The near-linear time algorithm is very efficient

- The presentation is very clear

**Weaknesses:**

- Technical contribution seems to be limited, since almost all steps are using known/standard techniques. However, I do find the combination of them natural and elegant

- The experiments did not compare with any other baselines

- The experiments did not use real datasets and only small-scale synthesized dataset is used

**Questions:**

- In the statement of Lemma 14, epsilon only appears in the running time of CountEigenvalues, and I don’t see how it affects the accuracy. Actually, I think your CountEigenvalue has this epsilon as input. I’m not saying the proof has a problem; I’m just saying the statement loos strange.


- Section 3.3, in the description of the main algorithm, do you also need to specify the epsilon for CountEigenvalues? In this context, I think you need to say epsilon is an input parameter?

- Actually, Theorem 6 does not have the parameter epsilon, whereas CountEigenvalues need a parameter epsilon. However, how this epsilon is picked in order to prove Theorem 6 is not clearly discussed. Does it depend on the universal constant C? If so, then maybe it’s good to state this dependence, since perhaps your algorithm is efficient even when C is not constant.

- In the statement of Theorem 6, k is defined as the number to satisfy \Upsilon_G(k) \geq C k. However, is k well defined, in particular, is the k that satisfies this unique? If it is not unique, then what’s the guarantee of your algorithm then?

**Details Of Ethics Concerns:**

None.

---

> ### Author Response · Authors · 2024-11-20
>
> We thank the reviewer for their careful reading and positive evaluation on our work. Here we respond to the concerns raised from the report.
>
> >**Question 1:** In the statement of Lemma 14, epsilon only appears in the running time of CountEigenvalues, and I don’t see how it affects the accuracy. Actually, I think your CountEigenvalue has this epsilon as input. I’m not saying the proof has a problem; I’m just saying the statement loos strange.
>
> **Response:** Thank you for pointing out this. In our updated submission, we re-stated Lemma 14 with respect to $\epsilon$ to make the statement more formal.
>
> >**Question 2:** Section 3.3, in the description of the main algorithm, do you also need to specify the epsilon for CountEigenvalues? In this context, I think you need to say epsilon is an input parameter?
>
> **Response:** Thank you for pointing out this. Yes, $\epsilon$ is part of the input, and we have updated it in the current version of our submission.
>
> >**Question 3:** Actually, Theorem 6 does not have the parameter epsilon, whereas CountEigenvalues need a parameter epsilon. However, how this epsilon is picked in order to prove Theorem 6 is not clearly discussed. Does it depend on the universal constant C? If so, then maybe it’s good to state this dependence, since perhaps your algorithm is efficient even when C is not constant.
>
> **Response:**  Thank you for pointing out this. To improve the clarity of the presentation, in the updated version of our submission we state the success probability as $1-o(1)$. Since $\epsilon$ is just a constant, its choice wouldn't asymptotically  influence the order of success probability. Moreover, it doesn't depend on the universal constant $C$. This independence of $C$ is important for our algorithm, as one cannot predict this value in advance.
>
> >**Question 4:** In the statement of Theorem 6, $k$ is defined as the number to satisfy $\Upsilon_G(k) \geq C k$. However, is $k$ well defined, in particular, is the $k$ that satisfies this unique? If it is not unique, then what’s the guarantee of your algorithm then?
>
> **Response:** The number of clusters k is defined as the minimum value for which there is a gap between $\lambda_k$ and $\lambda_{k+1}$; see the survey by von Luxburg. Hence, $k$ is uniquely defined.
>
> Finally, regarding the weakness, we remark that our algorithm is the *first* nearly-linear time algorithm for this problem, and our experiment is mainly to demonstrate that our algorithm runs in nearly-linear time in practice; we also believe that our experiments on small-scale synthesized dataset suffice for this purpose. If the review think that there is other specific baseline algorithms that we should compare with, we're happy to conduct more experiments.

---

> > ### Comment · Reviewer_ySsE · 2024-11-22
> >
> > Thanks for the response.
> >
> > For the uniqueness of k, you mentioned “the number of clusters k is defined as the *minimum* value to…”. I didn’t find this claim in your manuscript; instead, it is only saying k satisfies certain inequality (around Theorem 6), and the minimum is not mentioned. Are you saying that von Luxburg shows the definition mentioned in your manuscript is equivalent to what you just said about minimum?
> >
> > For the baselines, I’m not sure about concrete ones, but perhaps one could try some naive heuristics etc. Or maybe one could (binary/doubling?) search k, by utilizing an (existing) clustering algorithm which assumes k is given, and then look at how good the clustering looks like.

---

> > > ### Author Response · Authors · 2024-11-22
> > >
> > > Thank you for carefully reading our response.
> > >
> > > > For the uniqueness of k, you mentioned “the number of clusters k is defined as the minimum value to…”. I didn’t find this claim in your manuscript; instead, it is only saying k satisfies certain inequality (around Theorem 6), and the minimum is not mentioned. Are you saying that von Luxburg shows the definition mentioned in your manuscript is equivalent to what you just said about minimum?
> > >
> > > When determining the number of clusters in practice, people always choose the minimum $k$ satisfying this condition. von Luxburg shows that the number of clusters $k$ is the minimum value $k$ for which there is a large gap between $\lambda_{k}(N_G)$ and $\lambda_{k+1}(N_G)$, which is equivalent to the minimum value of $k$ satisfying our condition. We should have pointed out the minimum value $k$ in the manuscript, but this doesn't affect the correctness of our work.
> > >
> > > >For the baselines, I’m not sure about concrete ones, but perhaps one could try some naive heuristics etc. Or maybe one could (binary/doubling?) search k, by utilizing an (existing) clustering algorithm which assumes k is given, and then look at how good the clustering looks like.
> > >
> > > We didn't try some naive heuristic since searching the right value of $k$ for spectral clustering would require us to compute the eigenvalues and eigenvectors of the input graph, which takes $O(n^3)$ time. Hence, it's clear that they are not the potential competitors of our algorithm.
> > >
> > > Just let us know if you have more questions on our submission.

---

> > > > ### Comment · Reviewer_ySsE · 2024-11-22
> > > >
> > > > Thanks for the clarification. I don’t have further questions for the time being, and I would like to keep my rating for the paper.

---

### Official Review · Reviewer_9Tu9 · 2024-11-02

**Soundness:** 2
**Presentation:** 2
**Contribution:** 3
**Rating:** 5
**Confidence:** 3

**Summary:**

The paper introduces a novel algorithm designed to determine the number of clusters (k) in a graph based on the eigen-gap heuristic, which historically requires high computational effort due to its dependence on calculating all eigenvalues of the graph's normalized adjacency matrix. This research addresses one of the bottleneck of spectral clustering, presenting a new methodology that computes k in nearly-linear time. The algorithm, with high probability, accurately computes the number of clusters by analyzing the normalized adjacency matrix of the graph through a sparse representation.
Central to this approach is the strategic use of orthogonal polynomials, which facilitate an efficient approximation of the graph's spectral properties. This method avoids the computationally expensive task of direct eigenvalue computation by instead approximating the trace of the graph normalized adjacency matrix.

**Strengths:**

The strategy of sparsifying the graph and using orthogonal polynomials to efficiently approximate the trace seems new and constitute a nice contribution.

**Weaknesses:**

1) The effectiveness of the algorithm heavily relies on specific assumptions about the graph's structure, particularly the presence of well-defined clusters and clear spectral gaps.
While the algorithm presented in the paper represents a substantial theoretical advancement in the field of graph clustering, its practical applicability might be limited by its stringent reliance on specific graph conditions,. A critical oversight is the lack of a mechanism within the algorithm to pre-assess whether a given dataset meets these conditions. Without preliminary testing to verify these prerequisites, users might apply the algorithm to unsuitable datasets, leading to poor performance or invalid clustering results. Incorporating a diagnostic test as an initial step in the algorithm could significantly enhance its utility. This modification would make the algorithm more robust and adaptable, extending its relevance and effectiveness across a broader range of practical scenarios where data conditions are not ideal or well-understood in advance.

2) The paper may not provide extensive comparative analysis with other state-of-the-art clustering algorithms (with model selection), particularly those that might use different approaches such as density-based clustering or machine learning models that do not rely on spectral properties. This limits understanding of where the presented algorithm stands in the broader landscape of model selection techniques.

**Questions:**

1) Given the critical reliance of the algorithm on specific structural conditions within the graph, could the authors elaborate on any potential methods or strategies to integrate a diagnostic test within the algorithm to pre-assess whether these conditions are met in a given dataset? This addition would be crucial for ensuring the algorithm's adaptability and effectiveness.

2) Could you run your algo on difficult synthetic data and compare it to some state of the art and simple algorithms (like cross validation of some unsupervised criteria)?


Minor questions:
- Define the notation  h_{a,b}(M), T_i(M)...

- Are the Chebyschev polynomials used for a specific reason or other orthogonal polynomials could be considered as well?

- In Lemma 14, the probability obtained should not depend also on epsilon?

- It could maybe more insightful to state the main result with a probability that depend on the parameters of the problem (rather than 9/10).

---

> ### Author Response · Authors · 2024-11-20
>
> We thank the reviewer for their careful reading and insightful comments on our work. Here we response to the points raised from the report.
>
> >  **Question 1:** "Given the critical reliance of the algorithm on specific structural conditions within the graph, could the authors elaborate on any potential methods or strategies to integrate a diagnostic test within the algorithm to pre-assess whether these conditions are met in a given dataset? This addition would be crucial for ensuring the algorithm's adaptability and effectiveness."
>
> **Response:** This is  a very interesting and natural question following our current result. To elaborate on a potential method, notice that one can effectively approximate the trace of $N_G^k$ for any $k$ in nearly-linear time, using the techniques presented in the paper. Since $\mathrm{tr}(N_G^k)=\sum_{i=1}^n \lambda_i^k(N_G)$, one could expect different distributions of $\{\mathrm{tr}(N_G^k)\}$ for graphs with the eigen-gap and the other case. We will formally explore this and similar approaches, and add necessary discussion in the next version of our paper.
>
>
> > **Question 2:** "Could you run your algo on difficult synthetic data and compare it to some state of the art and simple algorithms (like cross validation of some unsupervised criteria)?"
>
> **Response:** We aren't very sure which state of the art and simple algorithms you refer to, since to the best of knowledge our algorithm is the first nearly-linear time algorithm that finds the number of clusters. However, if you have specific algorithms in mind against which we should compare our algorithm, we'll be happy to run your suggested experiments.
>
> > **Minor Question 1:** "Define the notation h_{a,b}(M), T_i(M)..."
>
> **Response:**  We define function $h_{a,b}: \mathbb{R} \rightarrow \mathbb{R}$ on Lines 211 -- 213, and Lines 095 -- 096 shows that how to generalise any function $f:\mathbb{R} \rightarrow \mathbb{R}$ to $f: \mathbb{R}^{n\times n} \rightarrow \mathbb{R}$. Here $f_{ab}$ applies to the eigenvalues of $M$.  We defined    the recursive definition of Chebyshev polynomials of first kind on Lines 110 -- 113. If we apply this recursive definition on matrix $M$, it becomes $T_0(M)=I, T_1(M)=M$, and $T_k(M)=2M\cdot T_{k-1}(M) - T_{k-2}(M)$ for any $k\geq 2$.
>
> > **Minor Question 2:** "Are the Chebyschev polynomials used for a specific reason or other orthogonal polynomials could be considered as well?"
>
> **Response:** This is a very interesting question. Here we apply the Chebyshev polynomials for the following reasons: (1)
>   Optimal Approximation: Chebyshev polynomials are widely regarded as the best choice for approximating functions on \([-1, 1]\). Since the eigenvalues of the normalized adjacency matrix also lie within this interval, Chebyshev polynomials are particularly well-suited for approximating the spectral density function \(s\). (2) Ease of Implementation: Chebyshev polynomials are defined recursively, making them computationally efficient and straightforward to implement compared to other orthogonal polynomials.
> We think other orthogonal polynomials with the two properties could be used in our problem as well.
>
> > **Minor Question 3:** "In Lemma 14, the probability obtained should not depend also on epsilon?"
>
> **Response:**  Thank you for pointing out this, and the probability should depend on $\epsilon$. We have updated  Lemma 14 to reflect this dependence.
>
> > **Minor Question 4:** "
> It could maybe more insightful to state the main result with a probability that depend on the parameters of the problem (rather than 9/10)."
>
>  **Response:** Thank you for pointing out this. In the updated version of our submission, we state that the success probability is $1-o(1)$. We chose not to write the success probability with respect to $\epsilon$ since there are several $\log$ term when applying the union bound over the failure probabilities and it's uncommon to hide these terms in $\tilde{O}(\cdot)$ when writing the success probability.

---

> > ### Comment · Reviewer_9Tu9 · 2024-11-21
> >
> > Question 2: I meant running special clustering with cross-validation using intra-cluster variance versus global variance criterion?
> > I guess people use many of such cross validation techniques in practice when the number of clusters is unknown...

---

> > > ### Author Response · Authors · 2024-11-22
> > >
> > > Thank you for the clarification and valuable suggestion. We will add more experimental results and discussion on the use of spectral clustering with cross-valuation to determine the number of clusters in the next version of our paper. However, to the best of our knowledge, the running time of these algorithms is much higher than the nearly-linear time complexity of our proposed approach.

---

> ### Comment · Reviewer_9Tu9 · 2024-11-29
> **Non-convincing responses**
>
> I was not reassured by the answers to some questions. Ill lower my grade as I believe the paper is definitely interesting but suffers so far from a few technical problems to be resolved...

---

### Author Response · Authors · 2024-11-25

Dear reviewers, with the discussion period ending soon, we’d really value your feedback on our responses - let us know if there’s anything else we should clarify. Thank you once more for your time.

---

### Meta-Review · Area_Chair_X51C · 2024-12-17

**Metareview:**

This paper proposes a novel algorithm for determining the number of clusters in a graph based on the eigen-gap heuristic.  The key contribution is a nearly-linear time algorithm that leverages orthogonal polynomials and sparse matrix representations to efficiently approximate the trace of the graph's normalized adjacency matrix, thereby avoiding the computationally expensive task of computing all eigenvalues. This addresses a significant bottleneck in applying the eigen-gap heuristic for spectral clustering.

Reviewers find the proposed algorithm and its theoretical analysis to be interesting and a step in the right direction for improving the efficiency of spectral clustering. The nearly-linear time complexity is a significant improvement over traditional methods.

However, reviewers also raise some concerns:

- Technical Issues in Proof: One reviewer identified potential technical issues in the proof that were not fully addressed by the authors during the rebuttal phase. This raises concerns about the correctness of the theoretical claims.
- Limited Experimental Evaluation: The experiments lack comparisons with other baseline methods for determining the number of clusters. This makes it difficult to assess the practical advantages of the proposed algorithm.
- Clarity of Proofs: The proofs could benefit from careful rewriting to improve their accessibility and clarity for a broader audience.

Recommendation:

While the paper presents a promising approach for determining the number of clusters in a graph, the reviewers agree that it does not meet the bar for acceptance at ICLR in its current form.

**Additional Comments On Reviewer Discussion:**

The discussion was interesting and an issue about a proof was identified

---

### Decision · Program_Chairs · 2025-01-22

Reject